chemical physics/green chemistry/materials science

agro-based ACs, pulping as synergistic treatment, pulp characteristics, surface area ($S_{BET}$), adsorption capacities measurements

**Author for correspondence:**
Altaf H. Basta
e-mail: altaf_basta2004@yahoo.com

This article has been edited by the Royal Society of Chemistry, including the commissioning, peer review process and editorial aspects up to the point of acceptance.

# Role of pulping process as synergistic treatment on performance of agro-based activated carbons

## Altaf H. Basta[1], Vivian F. Lotfy[1] and Philippe Trens[2]

[1]Department of Cellulose and Paper, National Research Centre, 33 El Bohouth Street, Dokki, Giza PO 12622, Egypt
[2]Institut Charles Gerhardt des Matériaux de Montpellier, MACS/CNRS/ENSCM/UM, 240 avenue du Pr Emile Jeanbrau, 34296 Montpellier Cedex 5, France

AHB, 0000-0003-1876-4378

To recommend the beneficial effect of the pulping process on enhancing agro-wastes as precursors for the production of high-performance activated carbons (ACs), different pulping methods (alkali, sulfite and neutral sulfite) were applied on two available Egyptian agriculture by-products (rice straw and sugar cane bagasse), using the one-step pyrolysis method and $H_3PO_4$ activating agent. The adsorption performance of the different prepared ACs was evaluated in terms of Iodine Numbers and their sorption properties for removing the methylene blue (MB) from aqueous solutions. The corresponding sorption processes were also analysed using Lagergren first order, pseudo-second order and intraparticle diffusion models. Data revealed that the applied pulping conditions were effective for removing the non-cellulosic constituents of agro-residues. This was demonstrated by the hydrogen/carbon and oxygen/carbon ratios, thermal stability and IR-measurements of the final pulps. These data were effective on the particular sorption properties of RS and SCB-based ACs. Interestingly, the pulping process is a profound modification of the SCB-based fibres, on which it induced a clear increase of the specific surface areas of the corresponding ACs even though they had an impact on the sorption of MB and iodine. These values are superior to the reported data on agro-based ACs with $H_3PO_4$ activators. Pulping processes therefore play a dual role in the sorption properties of ACs. The first important role is the impact on the specific surface areas and the second impact is a profound modification of the surface chemistry of the ACs. Therefore, SCB-based ACs can be seen as an economical breakthrough product, and an alternative to the high-cost commercial ACs for the purification of industrial wastewaters.

# 1. Introduction

Activated carbons (AC) are still highly regarded materials for a variety of applications, such as adsorbents for toxic materials and pollutants [1–3], as well as in the storage of hydrogen and methane [4,5], electrolysis [6], catalysis [7] and many medical and pharmaceutical uses [8,9]. On the other hand, large amounts of lignocellulosic wastes are accumulated in many countries, and their utilization instead of natural wood in engineered wood and other lignocellulosic composites and other value-added products [10–17] is still limited. This is clearly due to the variable, and usually poor, quality of the wastes. The utilization of agricultural wastes/by-products in the production of ACs has attracted many researchers, as an economic way to dispose of such available wastes as a valuable product. Many agriculture wastes/by-products are used by researchers for AC preparation such as rice straw (RS) [18], wheat straw [19], rice husk (RH) [20], sugar cane bagasse (SCB) [21], tobacco stem [22], palm shells [23], date stones [24], olive stones [25] and coconut shells [26]. The process of converting these wastes/by-products into ACs is based on their pyrolysis in the presence of physical or chemical activators [27]. Usually, agro-based ACs exhibit low to moderate specific surface areas, especially when they are subjected to one-step pyrolysis. Many researchers are thus working on improving the textural properties of ACs from these wastes/by-products [28,29]. Many different physical pretreatments were performed in order to reach this goal. For instance, coconut shell underwent a hydrothermal pretreatment in the presence of hydrogen peroxide as oxidizing agent [28]. Also, chemical pretreatments were applied, for example, by using sulfuric acid followed by an alkaline solution in the same batch [29]. The alkaline solution was also used as a post treatment after sulfuric acid treatment [30]. Using mechanical and alkali treatment, we have already succeeded in providing enhanced RS-based ACs, using one- or two-step KOH activation processes, the specific surface areas reaching 657 $m^2 g^{-1}$ and 1917 $m^2 g^{-1}$, respectively [31]. Chemical activation of RS using $H_3PO_4$ and a one-stage pyrolysis provided ACs with specific surface area lower than 500 $m^2 g^{-1}$, which is too low to comply with commercial ACs. In our former studies, we succeeded in improving the agro- or lignocellulosic products via physical, alkaline and biological pretreatments. We used organo-metallic compounds as additives and we provided xerogel-based ACs from different aldehydes such as HCHO-scavenger for wood adhesive [32–35]. In continuation, the objective of the present work was to evaluate the possibility of enhancing the most available agro-wastes (SCB & RS) as precursors for the production of ACs, via introducing $HSO_3$ groups together with changing the ratio of cellulose/hemicellulose/lignin. To achieve this objective, three pulping processes were applied using soda (SH), sulfite (SS) and sulfite–carbonate mixture (mix SS-SC) (neutral), followed by activation using the phosphoric acid. The prepared ACs were characterized via nitrogen adsorption experiments, thermal analysis and IR spectroscopy. The sorption properties of the ACs were assessed by performing iodine and methylene blue (MB) sorption from the liquid phase.

# 2. Experimental set-up

## 2.1. Materials

Sodium hydroxide, sodium sulfite and sodium carbonate used as reagents in the pulping process were purchased from El-Nasr pharmaceutical chemical Co, (ADWIC). Two Egyptian agriculture wastes (rice straw (RS) and sugar cane bagasse (SCB)) were used as precursors for active carbon preparation. Phosphoric acid (Farbwerke Hoechst AG, purity 85%) and MB (Alfa Chemicals Co, purity 98%) were used for activation and sorption performance assessment, respectively.

## 2.2. Preparation of agro-based precursors

The AC precursors RS and SCB were exposed to three different pulping processes to study their effect on the properties of the prepared AC in terms of sorption capabilities. The un-pulped RS and SCB fibres were labelled as [RS-cont.] and [B-cont.], respectively. The first process is an alkaline pulping using sodium hydroxide, equivalent to 6.55% $Na_2O$. The corresponding materials were labelled as [RS-SH] and [B-SH], respectively. The second process is an acidic pulping using sodium sulfite, equivalent to 6.55% $Na_2O$. The corresponding materials were labelled [RS-SS] and [B-SS], respectively. The third process is a neutral pulping using a mixture of sodium sulfite with sodium carbonate with mass ratio 4 : 1, equivalent to 6.55% $Na_2O$. The corresponding materials were named [RS-SC] and [B-SC], respectively. All pulping processes took place in autoclaves with liquor ratio of 5 : 1 at a temperature

of 140°C for 2 h. The fibres were further subjected to vigorous washing with distilled water. In a last stage, the materials were treated with a 10% aqueous solution of acetic acid and further washed by distilled water until neutralization.

All the materials were activated using phosphoric acid in ratio to pulp 3 : 1, followed by pyrolysis in a horizontal tubular furnace at 450°C for 60 min [34]. The obtained carbons were washed with distilled water until neutralization and dried in oven at 105°C.

## 2.3. Characterization of agro-based fibres and theirs activated carbons

### 2.3.1. Fibre characterization

#### 2.3.1.1. Chemical and elemental analyses

The chemical constituents of the fibres (lignin, holocellulose, α-cellulose and hemicellulose) were estimated according to the standard methods reported in references [36–38]. Their silica contents (as ash) were estimated after full oxidation of the materials in a muffle furnace in air at 800°C for 45 min.

The elemental analyses of the materials were determined using a Vario Elementar (Germany) elemental analyser (C, N, H and S). The mass fraction of oxygen was deduced from the calculation of the H/C and O/C ratios.

#### 2.3.1.2. Thermal analyses

Thermo-gravimetric analyses (TGA) of the precursors (un- and pulped fibres) were performed using a Perkin–Elmer Thermal Analysis Controller AC7/DX TGA7. The analyses were performed using a heating rate of 10°C/min and a nitrogen flow rate of 50 ml min$^{-1}$.

#### 2.3.1.3. FTIR spectra analysis

Infrared spectra were recorded with a Nicolet Nexus 670 spectrometer equipped with a Deuterated Triglycine Sulfate detector. The samples were mixed with KBr and pressed as tablets. The absorbance spectra were recorded in the region from 4000 down to 400 cm$^{-1}$. The influence of the pulping processes was clearly evidenced from the appearance of specific functional groups in the absorbance infrared spectra of the different precursors.

### 2.3.2. Adsorption experiments

#### 2.3.2.1. Nitrogen adsorption

The textural characterization of the AC samples prepared from un- and pulped fibres was carried out by nitrogen sorption performed at 77 K using a Nova 3200 Quantachrome Instrument. The samples were degassed in an oven at 250°C for 24 h. The adsorption isotherms were analysed in terms of textural properties (specific areas' pore volume and pore size distribution if useful).

#### 2.3.2.2. Iodine number

The Iodine Number is reported as the most fundamental parameter used to characterize AC performance. It was measured according to the procedure established by ASTM (D46-07-94) [39]. The equations allowing for the determination of the Iodine Number can be summarized according to the following:

$$\text{Iodine Number (mg g}^{-1}) = C \times \text{conversion factor} \qquad (2.1)$$

$$\text{Conversion factor} \ = \frac{40 \ \times \text{Mol wt. of iodine} \times \text{iodine normality}}{\text{carbon wt.} \times \text{blank reading}} \qquad (2.2)$$

and    $C = \text{blank reading} - \text{volume of sodium thiosulfate consumed after adsorption.}$ \qquad (2.3)

#### 2.3.2.3. Batch adsorption equilibrium and kinetic studies of methylene blue adsorption

Six concentrations ranging from 100 to 600 mg l$^{-1}$ of MB solution were added to the investigated AC powders with a constant AC/liquor wt. ratio of 400. The suspensions were kept in a shaker at a fixed

temperature of 30°C for 24 h. This duration has been determined as long enough to ensure a thermodynamic equilibrium between adsorbed MB species and the agro-based AC surfaces. After equilibrium, the MB solutions/AC mixtures were filtered, and the depleted solutions containing some MB were quantified by UV–Visible spectrophotometry (unico™UV-2000 spectrophotometer) working at a fixed wavelength of 662 nm [40].

The MB adsorption capacity at equilibrium, $Q_e$ (mg/g), was calculated using the following equation:

$$Q_e = \frac{(C_o - C_e) \times V}{W}, \tag{2.4}$$

where $C_o$ and $C_e$ (mg l$^{-1}$) denotes the liquid-phase concentration of MB at initial and equilibrium, respectively; $V$ (L), volume of the MB solution; $W$ ($g$), weight of the agro-based AC.

*Batch equilibrium studies.* The adsorption isotherm data were fitted using some classical adsorption isotherm models. This represents an important step to find the suitable model that can be used for AC design purposes. Langmuir and Freundlich isotherms are the most common ones. The Langmuir theory is valid for a monolayer adsorption onto a surface containing a finite number of identical sites. The linear form of the Langmuir isotherm equation is expressed as [41]

$$\frac{C_e}{q_e} = \frac{1}{bq_m} + \frac{C_e}{q_m}, \tag{2.5}$$

where $q_e$ is the amount adsorbed at equilibrium, $C_e$ is the equilibrium concentration of the adsorbate (MB), $q_m$ (mg g$^{-1}$) is the maximum adsorption capacity and b is the binding constant which is related to the enthalpy of adsorption.

The Freundlich adsorption isotherm model is valid for heterogeneous surfaces. The linear form of the Freundlich model is generally represented as follows [42]:

$$\log [q_e] = \log K_F + \frac{1}{n}\log [C_e], \tag{2.6}$$

where $K_F$ and $n$ are Freundlich constants, $n$ giving an indication of how favourable the adsorption process is and $K_F$ (mg g$^{-1}$ (l mg$^{-1}$)$^n$) is the adsorption capacity of the adsorbent.

The applicability of the different adsorption isotherm equations is judged by comparing the obtained correlation coefficients. In the case of the linear form of Langmuir's isotherm model, $C_e/q_e$ has to be plotted against $C_e$. If a straight line is obtained, it is indicative of an adsorption process verifying the Langmuir model's hypotheses, the slope of the straight line being $1/Q_m$. In the case of the Freundlich isotherm model, by plotting $\log q_e$ against $\log C_e$ a straight line with slope $1/n$ should be obtained for verifying the Freundlich model's hypotheses. In the Langmuir approach, the constant $b$ and the saturation capacity $Q_m$ were calculated in the case of all adsorption isotherms. The Freundlich constants $K_F$ and $n$ were also calculated. The essential characteristics of the Langmuir adsorption isotherm can be expressed in terms of a dimensionless equilibrium parameter ($R_L$), which is defined by

$$R_L = \frac{1}{1 + bC_m}, \tag{2.7}$$

where $b$ is the Langmuir constant and $C_m$ is the dye concentration (mg l$^{-1}$) corresponding to the sorbent saturation. The value of $R_L$ indicates the type of isotherm to be either unfavourable ($R_L > 1$), linear ($R_L = 1$), favourable ($0 < R_L < 1$) or irreversible ($R_L = 0$). All these results are gathered in table 6.

*Batch kinetic studies.* The kinetics of adsorption of MB on RS-ACs and B-ACs can be studied by applying the Lagergren first order, pseudo-second order and intraparticle diffusion models. These rate equations have been most widely used for the adsorption of an adsorbate from an aqueous solution. They are expressed by the equations found in [43–45]

| kinetic model | linear form | plots | ref. |
|---|---|---|---|
| Lagergren first order | $\mathrm{Ln}(q_e - q_t) = \mathrm{Ln}q_e - k_1 t$ | $\ln(q_e - q_t)$ versus $t$ | [43] |
| pseudo-second order | $\dfrac{t}{q_t} = \left[\dfrac{1}{k_2 q_e^2}\right] + \dfrac{1}{q_e}t$ | $t/q_e$ versus $t$ | [44] |
| intraparticle diffusion | $q_t = k_{id} \times t^{1/2} + C$ | $q_t$ versus $t^{1/2}$ | [45] |

where $q_e$ and $q_t$ are the amount of dye adsorbed per unit mass of the adsorbent (in mg g$^{-1}$) at equilibrium time and time $t$, respectively, $k_x$ are the rate constants, $C$ is the intraparticle diffusion constant.

### 2.3.2.4. Scanning electron microscope

The morphology of the investigated ACs was examined by scanning electron microscopy (SEM). The samples were exposed to gold coating (Edwards Sputter Coater, UK) using a quanta FEG250 system running at 20 kV.

# 3. Results and discussion

## 3.1. Evidence of changing the constituents of agro-wastes versus pulping processes

### 3.1.1. Chemical and elemental analyses

Table 1 summarizes the composition of raw RS and SCB, and their pulp fibres, in terms of the main chemical constituents (α-cellulose, hemicellulose, lignin and ash). The data show that, upon any type of pulping, there is reduction of both ash and lignin (as klason lignin) contents. At the same time, pulping processes increase both α-cellulose and hemicellulose (as pentosans). In general terms, pulping processes have a great effect on removing the wax and silica, leading to a decrease of the ash content [33]. In the case of RS fibres, the soda pulping is considered as the most effective treatment for removing the silica, as shown by a decrease in ash content from 18.4 to 14.8%. On the other hand, the neutral sulfite pulping leading to RS-SC provides the lowest reduction in ash content (from 18.4 to 17.5%). The same trend is observed with SCB fibres, whereas the ash reduced from 4.7 to 1.3% for soda pulping and to 1.5 and 1.8% with other pulping processes. It can also be noted that pulping of both RS and SCB leads to a general decrease of the klason lignin, the soda pulping being the most effective in the case of SCB fibres.

It must be emphasized that pulping processes has a profound effect on the chemical constituents of the produced pulps. Soda pulping provides the greatest removal of non-cellulosic components

**Table 1.** Chemical constituents of un- and different pulped RS and B fibres. The accuracy of the measurements is 0.1%.

| sample code | ash (%) | lignin (%) | holocellulose (%) | α-cellulose (%) | hemicellulose (%) |
|---|---|---|---|---|---|
| RS-cont. | 18.4 | 14.5 | 64.0 | 37.5 | 22.8 |
| RS-SH | 14.8 | 12.6 | 74.4 | 46.3 | 27.5 |
| RS-SS | 15.6 | 12.8 | 68.6 | 36.2 | 31.4 |
| RS-SC | 17.5 | 12.3 | 69.1 | 39.2 | 29.0 |
| B-cont. | 4.7 | 19.1 | 68.9 | 41.6 | 26.5 |
| B-SH | 1.3 | 14.0 | 76.3 | 52.6 | 23.0 |
| B-SS | 1.5 | 18.1 | 71.5 | 44.0 | 26.9 |
| B-SC | 1.8 | 15.9 | 72.5 | 48.8 | 23.1 |

**Table 2.** Elemental analysis of different un- and pulped RS and SCB fibres. The accuracy of the measurements is 0.01%.

| sample code | N% | C% | S% | H% | O% | H/C | O/C |
|---|---|---|---|---|---|---|---|
| RS-cont. | 0.61 | 36.70 | nil | 7.20 | 55.49 | 2.34 | 1.14 |
| RS-SH | 0.22 | 36.05 | nil | 8.20 | 55.53 | 2.71 | 1.16 |
| RS-SS | 0.40 | 35.43 | 0.21 | 7.80 | 56.17 | 2.62 | 1.19 |
| RS-SC | 0.36 | 36.21 | 0.15 | 8.80 | 54.48 | 2.90 | 1.13 |
| B-cont. | 0.29 | 44.00 | nil | 9.40 | 46.31 | 2.55 | 0.79 |
| B-SH | 0.19 | 41.88 | nil | 6.50 | 51.43 | 1.85 | 0.92 |
| B-SS | 0.28 | 43.96 | 0.62 | 6.80 | 48.34 | 1.84 | 0.83 |
| B-SC | 0.32 | 43.26 | 0.32 | 6.40 | 49.70 | 1.76 | 0.86 |

(lignin and hemicellulose). On the other hand, the lowest reduction in hemicellulose and lignin is observed when applying sulfite pulping. Indeed, in the case of RS-SH, RS-SS and RS-NP, the α-cellulose content increased from 37.5% to approximately 46%, 36% and 39%, respectively. In the case of SCB-pulps, this content increased from approximately 42% to 53%, 44%, approximately 49%, respectively. Sulfite and neutral sulfite pulping are less effective at reducing the lignin and pentosans content than soda pulping. The hemicellulose content is more difficult to interpret, as it increases in the case of RS fibres upon pulping (for RS-SC, from 23% to 31% and for RS-SS from 23% to 29%). Interestingly, in the case of SCB fibres the hemicellulose contents depends on the pulping process, soda and neutral sulfite pulping leading to a clear decrease of the hemicellulose content.

The elemental analyses of the different materials before and after pulping processes are gathered in table 2. Sulfite and neutral sulfite pulps included sulfur atoms, due to the introduction of the $SO_3H$ groups during the pulping process. The percentage of S % in the case of SCB-SS and SCB-SC (0.62% and 0.32) are higher than pulps from RS (0.21% and 0.15%). The ratio of hydrogen to carbon elements (H/C ratio) indicates the degree of aromaticity of the fibre structure [42,43]. For the RS-pulp, the H/C ratio strongly increased upon pulping for all processes. However, the O/C ratio remained pretty stable which means that pulping processes do not oxidize the RS fibres to a quantitative extent. However, in the case of SCB fibres, reverse trends can be observed. Indeed, for these materials, the H/C ratio is strongly decreased from 2.55 down to approximately 0.80, whereas the O/C ratio increased from 0.79 to approximately 0.85 on average. These results can be rationalized by considering that the role of pulping processes is the removal of the highest carbon content constituent (lignin), and an increase of the cellulose content.

### 3.1.2. Thermal analyses

Non-isothermal TGA was performed as a preliminary analysis to specify the temperature that could be applied for pyrolysis, as well as to view the changes in thermal stability due to constituents of RS and SCB, as a result of pulping processes (table 3 and figures 1 and 2). The thermal decomposition of hemicellulose and glycosidic linkages of cellulose can be seen at around 300°C. More precisely, the DTG peaks appear in the range of 320–360°C, which describes the decomposition of lignin [29]. In the case of the RS-pulp fibres, the onset temperatures of the degradation stage are higher for the pulped RS fibres (245–250°C) as compared with the raw RS (186°C). The highest onset temperature is found with the soda pulping fibre (RS-SH) (250.4°C). This indicates that pulping processes lead to the removal of low chain length components. The onset temperature location for degradation of pulp can be ascribed to the thermal stability of the pulps related to their constituents (lignin > cellulose > hemicellulose) (table 2).

**Table 3.** DTG/TGA peak analysis of different un- and pulped RS and SCB fibres. The temperature accuracy for the TGA measurements is 0.1°C.

| sample code | $T_i$ (°C) | $T_F$ (°C) | temp peak (°C) |
| --- | --- | --- | --- |
| RS-cont. | 186.1 | 385.0 | 320.1 |
| RS-SH | 250.4 | 385.1 | 350.5 |
| RS-SS | 245.0 | 386.3 | 351.3 |
| RS-SC | 245.0 | 390.1 | 350.6 |
| B-cont. | 220.1 | 385.2 | 349.8 |
| B-SH | 250.4 | 388.4 | 350.4 |
| B-SS | 242.3 | 390.6 | 355.5 |
| B-SC | 236.5 | 385.9 | 347.1 |

Concerning the DTG peak locations (figures 1 and 2 and table 3), they appear at higher temperatures for the RS pulps, especially when produced from sulfite pulping (351.33°C), as compared with the control RS fibres (320.05°C). This may be related to the presence of some lignin as lignin sulfonate,

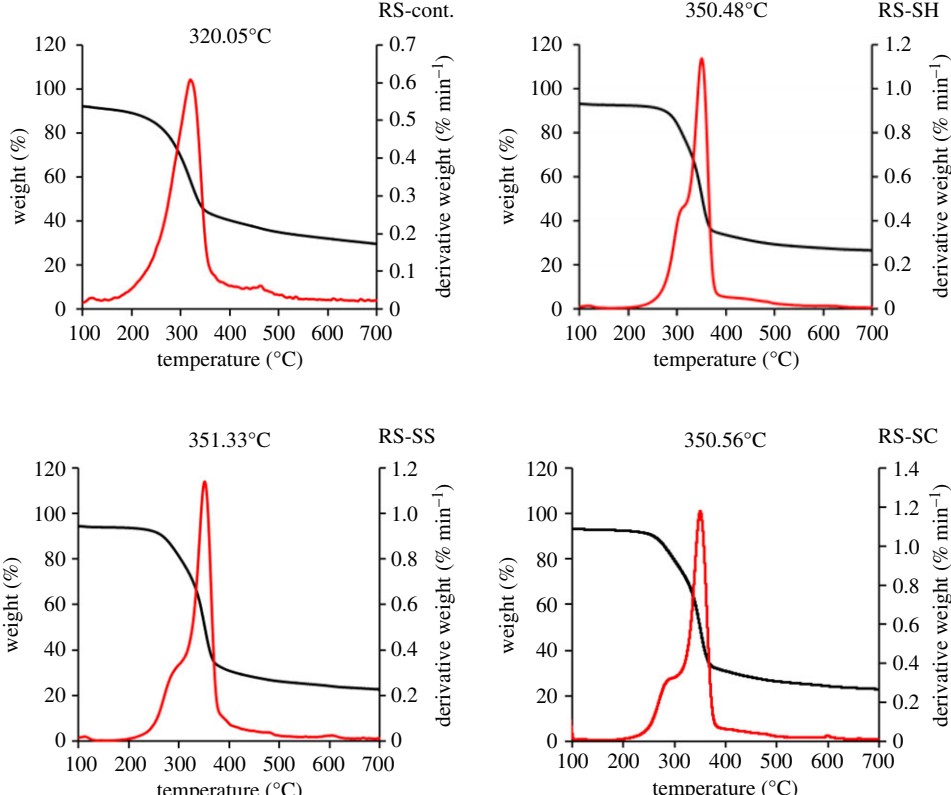

**Figure 1.** TGA/DTG analysis of different un- and pulped RS-fibres.

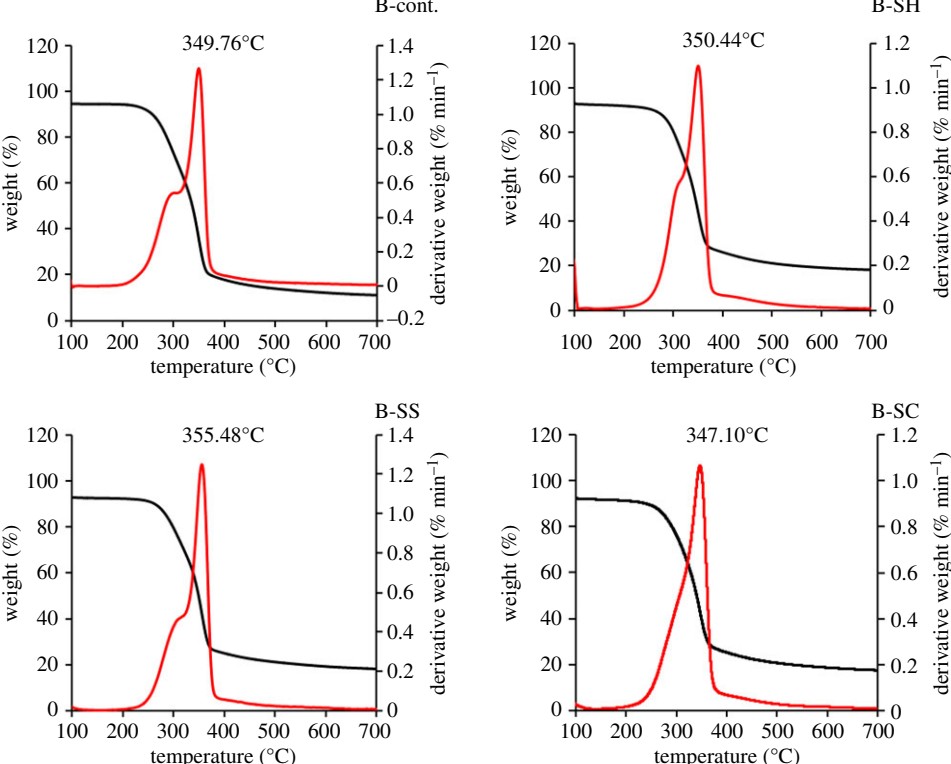

**Figure 2.** TGA/DTG analysis of different un- and pulped SCB-fibres.

which exhibits a relatively higher stability than lignin, together with the removal of low molecular weight components (waxes, resins and low chains hemicelluloses). Indeed, the remaining lignin content after the sulfite pulping process is 12.8%.

The same trend can be observed when focusing on the SCB-pulped fibres, as the onset temperature increased (236.5–250.4°C depending on the pulping process) as compared with control B fibre (approx. 220°C). Sulfite pulp has a higher DTG peak temperature than soda pulp, despite the fact that the latter pulp has a α-cellulose content of approximately 53%, as compared to 44% in the case of the former. This can be interpreted by considering that sulfite pulping has less effect on the removal of lignin than soda pulping (18% of lignin in B-SH versus 14% of lignin in B-RH). This difference can explain the difference in terms of thermal stability.

Besides, the TGA and DTG curves indicate that pyrolysis can be achieved for carbonization of the foregoing pulps in the temperature range 400–500°C.

### 3.1.3. FT-IR analysis agro-fibres versus pulping process

Figures 3 and 4 present the FT-IR spectra of RS and SCB and their pulps produced from the foregoing different pulping agents. The IR spectra are characterized by a strong band at about 3434 cm$^{-1}$ assigned to OH stretching and a low intensity band at 2927 cm$^{-1}$ assigned to CH asymmetric stretching. The small and sharp band appearing at 1630 cm$^{-1}$ can be assigned to C=O and C=C stretching, whereas the band at 1460 cm$^{-1}$ can be assigned to phenolic lignin OH bending. The sharp band appearing at approximately 1170 cm$^{-1}$ can be attributed to the ether group C–O–C linkage between the lignin to cellulose, hemicelluloses, even though it could also be attributed to the stretching of the Si–O–cellulose and Si–O–Si bonds [46] (table 4).

In the case of RS fibres, the OH stretching appearing at 3435 cm$^{-1}$ is shifted to lower wavenumbers (3385–3406 cm$^{-1}$) upon the three different pulping processes. This may be related to the removal of the wax and silica layer (i.e. ash content) from the RS surface. This is known to favour the intramolecular hydrogen bond formation between the chain fibres which shifts the band to lower wavenumbers [47].

The ratio of absorbencies at 3434 cm$^{-1}$ to that at 2927 cm$^{-1}$ indicates mean hydrogen bond strength (MHBS) [48]; while the ratio at 1430 cm$^{-1}$ to that at 900 cm$^{-1}$ indicates the crystallinity index (Cr. I.) [49]. The MHBS of pulped RS fibres is increased from 2.05 to 2.5 ± 0.1. This can be ascribed to the enhancement and reformation of intermolecular hydrogen bonds as a result of the breaking of the ether and β-glycosidic linkage between cellulose with lignin and hemicellulose chains during the pulping processes. The lower reduction of lignin or/and hemicellulose content by sulfite or neutral pulping is reflected by the higher Cr. I of RS-SS and SCB-SS. Indeed, they increased from 3.9 to 11.0 and from 10.7 to 17.9, respectively. Moreover, this indicates that the pulping processes make the fibres more ordered (crystalline). It can be deduced that the pulping processes likely help remove the unordered constituents in addition to the low molecular weight of hemicelluloses (amorphous holocelluloses).

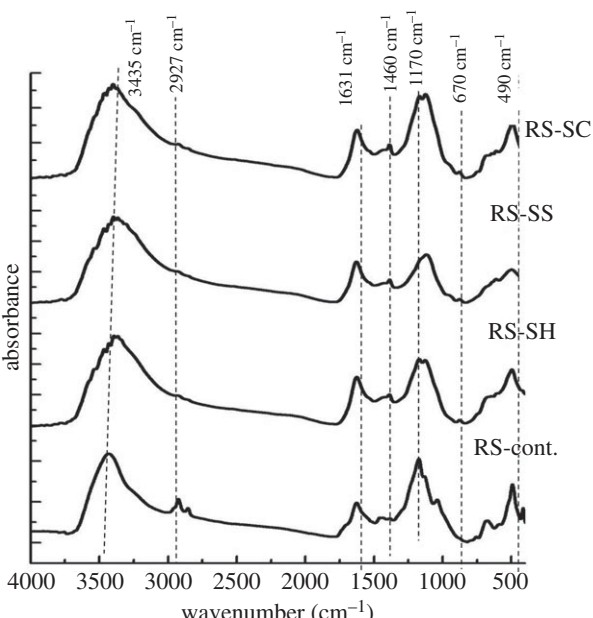

**Figure 3.** FTIR analysis of different un- and pulped RS-fibres.

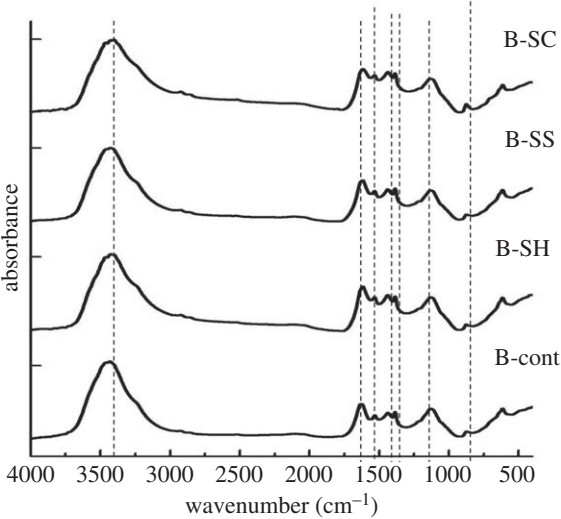

**Figure 4.** FTIR analysis of different un- and pulped SCB-fibres.

**Table 4.** IR characteristics of different un- and pulped RS and SCB fibres. The precision of the IR characteristics MHBS and Cr.I. is 0.01.

| sample code | MHBS ($A_{3434}/A_{2927}$) | Cr.I. ($A_{1430}/A_{900}$) |
|---|---|---|
| RS-cont. | 2.06 | 3.93 |
| RS-SH | 2.71 | 8.69 |
| RS-SS | 2.51 | 11.01 |
| RS-SC | 2.62 | 6.38 |
| B-cont. | 10.56 | 10.79 |
| B-SH | 5.15 | 13.48 |
| B-SS | 6.85 | 17.89 |
| B-SC | 3.54 | 17.03 |

## 3.2. Electron microscopy of ACs obtained after pulping processes

The morphology of ACs obtained from the different pulps is shown in figures 5 and 6. In the case of RS-ACs, figure 5 shows that the AC of un-pulped straw exhibits a heterogeneous surface, i.e. irregular shaped particles which are aggregated together. This is likely due to the presence of a wax/silica layer which forms cracks and crevices during the activation process using phosphoric acid. However, the pulped straw ACs, characterized by a partial removal of the wax/silica layer, appears highly porous, i.e. spherical shaped aggregates with a variety of randomly distributed macropore sizes. The same features appear with B-ACs (figure 6), as the pulped B-ACs exhibit a highly porous texture, which could be due to the partial removal of non-cellulosic compounds (lignin and hemicellulose). However, the entire textural properties of the ACs cannot be deduced using SEM only. Nitrogen adsorption was therefore performed to give deeper evidence of the textural properties of the differently prepared ACs (table 5).

## 3.3. Adsorption properties of activated carbons after pulping processes

### 3.3.1. Nitrogen adsorption at 77 K

The nitrogen adsorption isotherms performed on the ACs prepared from the un- and pulped RS and SCB fibres are presented in figure 7, and their textural features are collected in table 5. All the ACs exhibit the same general textural features. These materials are mostly microporous as seen by the very high nitrogen

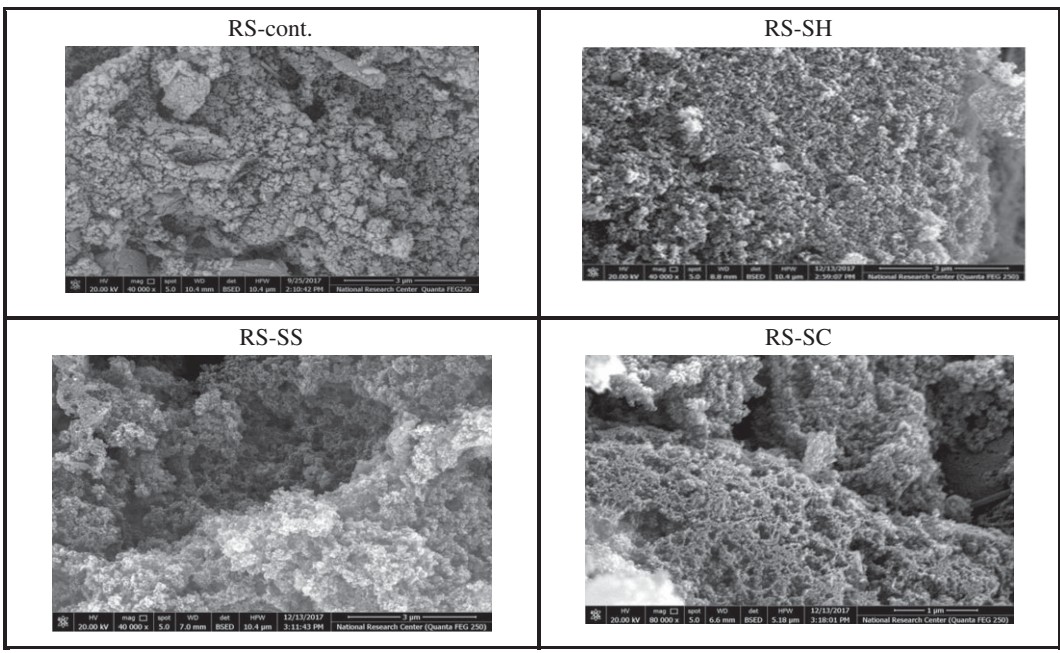

**Figure 5.** SEM of ACs prepared from different un- and pulped RS-fibres.

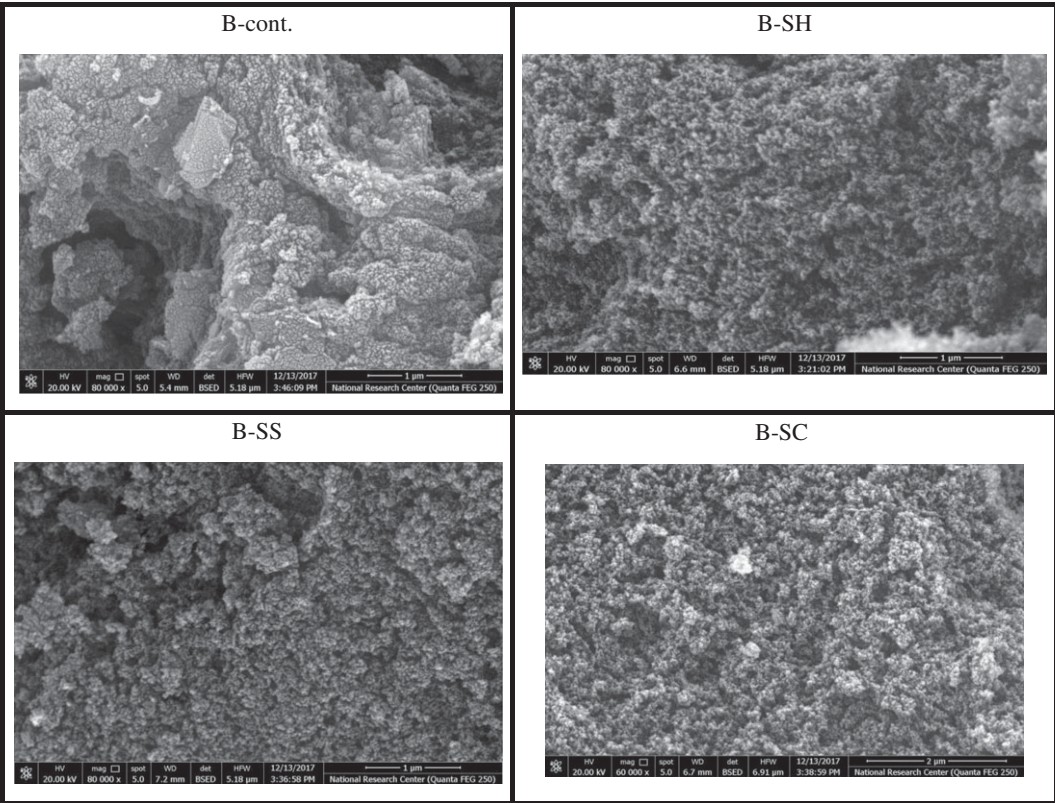

**Figure 6.** SEM of ACs prepared from different un- and pulped SCB fibres.

uptake at low relative pressure. This follows the type I of adsorption isotherms, according to the IUPAC classification. However, depending on the origin of the precursors, interesting differences can be seen. In the case of RS-based ACs, some heterogeneous mesoporosity can be distinguished, according to the steep slope in intermediate relative pressure range (between 0.3 and 0.8). This mesoporosity is present in the case of RS-based AC, but also in the case of the pulped RS-based ACs. It can also be noted that the pulping processes do not lead to an enhancement of the textural properties of the ACs

**Table 5.** Textural characterization of ACs prepared from different un- and pulped RS and SCB fibres.

| sample code | carbon yield % | $S_{BET}$ (m² g⁻¹) | $S_{mic}$ (m² g⁻¹) | $V_{T(0.95)}$ (cm³ g⁻¹) | $V_{mic}$ (cm³ g⁻¹) | $V_{mes}$ (cm³ g⁻¹) | $V_{mic}/V_T$ | pore radius (nm) |
|---|---|---|---|---|---|---|---|---|
| RS-cont. | 60.5 | 543.3 | 328.1 | 0.30 | 0.15 | 0.15 | 50.70 | 1.92 |
| RS-SH | 61.1 | 493.4 | 294.0 | 0.26 | 0.14 | 0.13 | 51.14 | 1.91 |
| RS-SS | 55.2 | 481.8 | 294.8 | 0.25 | 0.14 | 0.11 | 54.40 | 1.91 |
| RS-SC | 69.3 | 440.3 | 277.7 | 0.21 | 0.13 | 0.08 | 60.66 | 1.93 |
| B-cont. | 31.7 | 956.6 | 726.3 | 0.54 | 0.46 | 0.08 | 84.92 | 1.12 |
| B-SH | 38.4 | 1486.5 | 1025.8 | 0.57 | 0.47 | 0.10 | 82.14 | 1.91 |
| B-SS | 41.7 | 1093.8 | 861.9 | 0.56 | 0.41 | 0.15 | 73.31 | 1.92 |
| B-SC | 35.4 | 1431.6 | 981.85 | 0.569 | 0.45 | 0.12 | 78.91 | 1.91 |

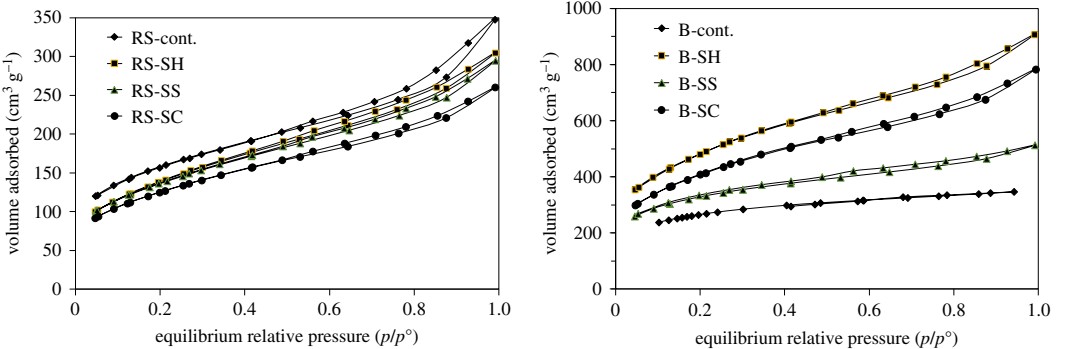

**Figure 7.** Adsorption isotherms of nitrogen on various ACs prepared from different un- and pulped RS and SCB-fibres performed 77 K.

produced. The equivalent specific surface areas are lower than in the case of AC obtained using unpulped RS. The worst case in terms of sorption properties seems to be the RS-SC as its textural specifications are lower than the other RS-based ACs (lower specific surface area and lower micropores volume, table 5). For example, the $S_{BET}$ of ACs prepared from RS-SH, RS-SS and RS-SC are decreased from 543.3 to 493.4, 481.8 and 440.3 m² g⁻¹, whereas the microporous volume decreased from 0.304 to 0.264, 0.25 and 0.211 cm³ g⁻¹, respectively The very thin hysteresis loops observed for all the RS-based ACs indicates that the aggregation of AC particles is very loose, which prevents the quantitative occurrence of an interparticular nitrogen adsorption.

When focusing on SCB-based ACs, different observations can be made. Firstly, the adsorption isotherm obtained with B-Cont already indicates that without any pulping process, B-Cont has better textural properties as compared to RS-cont. This is true in terms of adsorbed amounts at saturation, but also in terms of equivalent specific surface area. Furthermore, upon pulping processes, the textural properties of the corresponding ACs are clearly enhanced. In table 5, it can be seen that the surface area ($S_{BET}$) and total pore volume ($V_T$) of the prepared SCB-based ACs are significant affected by the pulping processes. Indeed, the pulping of SCB leads to a striking increase of both the $S_{BET}$ and $V_T$ of SCB-ACs as compared to the raw form. For example, the $S_{BET}$ of ACs prepared from B-SH, B-SS and B-SC is increased from 956.6 to 1486.5, 1093.8 and 1431.6 m² g⁻¹, respectively. At the same time, the total porous volume increases from 0.537 to 0.571, 0.562 and 0.569 cm³ g⁻¹, in the same sequence. These textural differences must have important consequences for the sorption properties of the prepared ACs.

### 3.3.2. Liquid-phase adsorption studies

The adsorption of iodine is shown in figure 8. In general terms, pulping processes enhance the sorption capacities of the obtained ACs, whatever the pulping type. Indeed, it can be noticed that the iodine

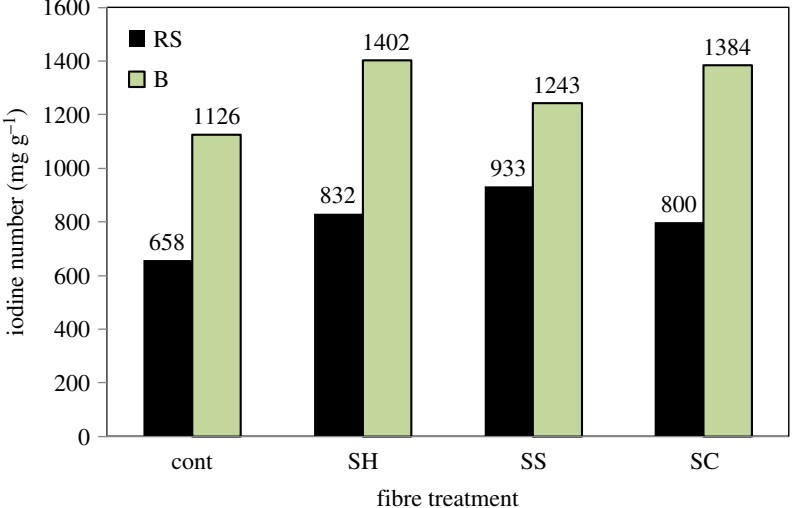

**Figure 8.** Iodine number of ACs prepared from different un- and pulped RS and SCB fibres at 25℃.

number is increased in the case of ACs prepared from pulped RS- and SCB-ACs, in comparison with raw (un-pulped) fibres. It is expected that pulped B-ACs have better sorption capacities as this property is usually related to the extent of specific surface area. Indeed, we demonstrated that pulping processes on SCB drastically increase the specific surface areas of the produced ACs. Iodine values go from 1126 mg g$^{-1}$ in the case of B-Cont AC to up to 1402 mg g$^{-1}$ for B-SH AC. However, the difference in terms of sorption capacities is not proportional to the extent of specific surface areas.

In the case of RS-based ACs, there is an increase of iodine sorption capacities observed with any of the pulping process considered as compared to RS-Cont AC (from 660 up to 850 mg g$^{-1}$ on average). Very interestingly, this increase occurs while the specific surface areas of the pulped RS-ACs are lower than in the case of RS AC. For the sake of comparison, it is interesting to note that the highest iodine number corresponding to RS-ACs and SCB-ACs exceeded the iodine number of xerogel-ACs (337–872 mg g$^{-1}$) based on different aliphatic aldehydes chains under the same activation conditions [34]. From these observations, it can be deduced that surface chemistry plays a primary role in the iodine sorption capacity of ACs, the different porosities required for quantitative sorption, facilitating the diffusion processes and therefore the kinetics of adsorption. The key roles of the pulping processes are therefore changes of constituents of agro-fibres as well as the redistribution of micro-meso pore volumes, changes of specific surface area, and each role dominates depending on the raw material studied. The observed favours iodine adsorption in the case of RS pulp-based ACs, despite the decrease in $S_{BET}$ which is probably ascribed to the increase in the micro pore volume with respect to the total pore volume (from 50 to 61%), as well as the *in situ* included silica in ACs. While the greatest specific surface area of SCB pulp-based materials is related to the promotion of their iodine sorption capacities as compared to that of un-pulped-fibres-based ACs.

We also investigated the efficiency of the prepared ACs to adsorb MB as shown in figure 9. All adsorption isotherms exhibit high slope at low concentration, indicative of high affinity sorption processes. To better understand the sorption process by the different ACs, we modelled the adsorption isotherms using two classical models. Table 6 summarizes the parameters values obtained from the two models and the corresponding correlation coefficients. It can be noted that the experimental data are better described by the Langmuir model, where it has better fit ($R^2$) than the Freundlich model. The correlation coefficient values ($R^2$) are approximately greater than 0.99. In addition, the dimensionless equilibrium parameter ($R_L$) is found between 0 and 1. This is an indication that the adsorption processes are favoured. The calculated data exhibit a maximum MB adsorption of 211.9 mg g$^{-1}$ corresponding to RS-SS-ACs and 403.2 mg g$^{-1}$ corresponding to SS-SH-ACs.

We also mentioned the meaning of the parameter $b$ in the Langmuir model, as related to the enthalpy of adsorption. It can be noticed that RS-Cont and B-Cont materials have similar $b$ values which suggests similar MB/AC interaction. After pulping, the $b$ parameters are very different, depending on the pulp employed. In the case of RS-based materials, pulping does not induce a significant change in the $b$ parameter. However, in the case of SCB-based materials, the $b$ parameter is strongly increased which suggests strong interaction between MB and the ACs.

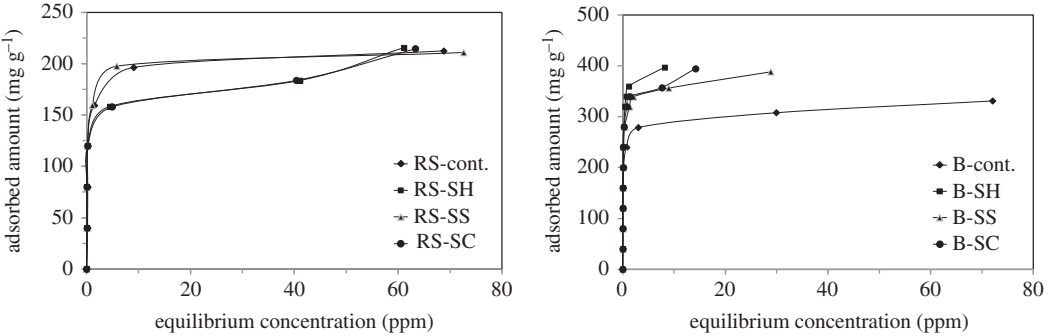

**Figure 9.** MB adsorption by ACs prepared from different un- and pulped RS and SCB fibres at 25℃.

**Table 6.** Langmuir and Freundlich isotherm parameters for adsorption of MB dye onto ACs prepared from different un- and pulped RS- and SCB-fibres.

| sample code | Langmuir isotherm | | | | | | Freundlich isotherm | | | |
|---|---|---|---|---|---|---|---|---|---|---|
| | slope $\times$ $10^{-3}$ | $Q_m^{-1}$ (mg g) | Inter. $\times$ $10^{-3}$ | $b$ | $R^2$ | $R_L \times$ $10^{-3}$ | $1/n =$ slop | Inter. | $K_f$ | $R^2$ |
| RS-cont. | 4.67 | 214.1 | 2.60 | 1.78 | 0.999 | 0.93 | 0.214 | 2.03 | 106.66 | 0.639 |
| RS-SH | 4.82 | 207.5 | 4.00 | 1.21 | 0.991 | 1.37 | 0.175 | 2.04 | 109.12 | 0.800 |
| RS-SS | 4.72 | 211.9 | 1.30 | 3.69 | 0.999 | 0.45 | 0.189 | 2.08 | 119.67 | 0.664 |
| RS-SC | 4.80 | 208.3 | 4.50 | 1.07 | 0.992 | 1.56 | 0.173 | 2.03 | 107.75 | 0.641 |
| B-cont. | 2.94 | 340.1 | 1.80 | 1.67 | 0.999 | 1.00 | 0.176 | 2.26 | 181.01 | 0.730 |
| B-SH | 2.48 | 403.2 | 0.30 | 7.29 | 0.999 | 0.23 | 0.325 | 2.51 | 321.66 | 0.603 |
| B-SS | 2.57 | 388.7 | 0.50 | 5.72 | 0.999 | 0.29 | 0.262 | 2.41 | 257.75 | 0.733 |
| B-SC | 2.56 | 390.2 | 0.40 | 6.57 | 0.998 | 0.25 | 0.261 | 2.43 | 271.02 | 0.673 |

The kinetics of MB adsorption onto the investigated ACs are studied using three classical models (Lagergren first order, pseudo-second order and intraparticle diffusion). The obtained kinetic parameters are summarized in table 7. As obtained from table 7, the correlation coefficient $R^2$ of the pseudo-second-order model (0.999–1) is higher than the other two models. Also, the computed adsorption capacity values ($q_e$) according to pseudo-second-order model are in agreement with experimental values ($Q_e$(exp)). From the foregoing results, it is expected that the adsorption of the MB on the surface of the investigated RS-ACs and SCB-ACs follows the pseudo-second-order mechanism which implies the interaction of MB with surface functionalities.

## 4. Conclusion

In continuation of our achievements in improving the surface adsorption of ACs from agro-wastes/ by-products, we examined the performance of ACs synthesized from RS and SCB versus pulping pretreatment processes. Our findings revealed that the pulping agent (soda, sulfite and sulfite– carbonate) plays a significant role in changing the chemical constituents of RS- and SCB-pulps. Alkaline, acidic and neutral pulping processes are more effective on bagasse-based ACs as compared to RS-based ACs, both in terms of surface area and adsorption capacities of iodine and MB. The alkaline pulping provides the greatest increase in surface area. Even though the sulfite and neutral sulfite pulping processes provided relatively lower $S_{BET}$, their adsorption capacities for MB (390 mg g$^{-1}$) approached the AC from alkaline pulp (403 mg g$^{-1}$). Based on a literature survey dealing with MB adsorption, I$_2$-values of different ACs from different agro-wastes, our pulping processes yield ACs with higher sorption capacities than what can be found in the literature (table 8).

**Table 7.** Lagergren first-order, pseudo-second-order kinetic model and intraparticle diffusion parameters for desorption of MB onto ACs prepared from different un- and pulped RS and SCB fibres.

| sample code | Lagergren first-order model | | | | | | pseudo-second-order | | | | | | intraparticle diffusion | | | |
| --- | --- | --- | --- | --- | --- | --- | --- | --- | --- | --- | --- | --- | --- | --- | --- | --- |
| | $Q_e$(exp) mg g$^{-1}$ | $K_1$ (h$^{-1}$) | $I$ | $q_e q$ (mg g$^{-1}$) | $R^2$ | SEE | $I \times 10^{-3}$ | $K_2$ (h$^{-1}$) | $S \times 10^{-3}$ | $q_e q$ (mg g$^{-1}$) | $R^2$ | SEE $\times 10^{-3}$ | $K_{id}$ | $C$ | $R^2$ | SEE |
| RS-cont. | 196.38 | 0.133 | 3.45 | 31.50 | 0.88 | 0.101 | 0.70 | 0.042 | 5.40 | 185.19 | 0.999 | 3.1 | 4.46 | 167.06 | 0.59 | 5.62 |
| RS-SH | 183.55 | 0.046 | 4.21 | 67.36 | 0.95 | 0.032 | 7.20 | 0.005 | 5.90 | 169.49 | 1 | 13.2 | 12.38 | 103.31 | 0.68 | 13.82 |
| RS-SS | 197.69 | 0.044 | 4.25 | 70.11 | 0.98 | 0.042 | 6.60 | 0.004 | 5.30 | 188.68 | 0.999 | 9.9 | 13.79 | 112.41 | 0.77 | 12.32 |
| RS-SC | 183.84 | 0.062 | 4.25 | 70.11 | 0.95 | 0.091 | 0.70 | 0.050 | 5.90 | 169.49 | 0.998 | 14.1 | 12.05 | 104.31 | 0.64 | 14.75 |
| B-cont. | 199.93 | 0.166 | 4.66 | 105.64 | 0.88 | 0.127 | 7.20 | 0.004 | 5.20 | 192.31 | 0.998 | 2.4 | 21.54 | 97.08 | 0.95 | 7.57 |
| B-SH | 199.97 | 0.264 | 2.87 | 17.64 | 0.94 | 0.109 | 0.60 | 0.042 | 5.00 | 200.00 | 1 | 0.1 | 3.10 | 186.36 | 0.83 | 2.25 |
| B-SS | 199.95 | 0.285 | 3.76 | 42.95 | 0.95 | 0.118 | 1.50 | 0.016 | 4.90 | 204.08 | 1 | 0.2 | 7.60 | 166.64 | 0.82 | 5.76 |
| B-SC | 199.95 | 0.169 | 3.11 | 22.42 | 0.94 | 0.171 | 0.80 | 0.031 | 5.00 | 200.00 | 1 | 0.2 | 3.73 | 183.3 | 0.95 | 1.32 |

**Table 8.** Comparing the adsorption behaviour of our present ACs with literature ACs from different agro-wastes and xerogels, using $H_3PO_4$ activating agent.

| adsorbent | Langmuir Ads. capacity (mg g$^{-1}$) | I-value (mg g$^{-1}$) | surface area (m$^2$ g$^{-1}$) | ref. |
|---|---|---|---|---|
| sugar-cane bagasse | 150–177 | 648–890 | 1075–1254 | [50] |
| *Thevetia peruviana* | 532 | 798 | 862 | [51] |
| orange peel | 41.9 | | 1090 | [52] |
| hazelnut husks | 204 | | 770 | [53] |
| rice straw | 215 | 855 | 967.72 | [54] |
| *Baslsamodendron caudatum* | | | 505 | [55] |
| wood waste | | | | |
| peanut hulls | 149 | 813 | 813 | [56] |
| rice straw | 198 mg g$^{-1}$ | 629 | | [57] |
| banana leaves | 19–48 | | 798–1228 | [58] |
| apple pulp | 283.8 | | 1103 | [59] |
| corncob | 18–29 | | | [60] |
| *Acacia (Vachellia seyal)* tree | — | 837 | 762 | [61] |
| our work | | | | |
| bagasse pulp-based ACs | 388–403 | 1243–1402 | 1094–148 | |
| RS pulp-based ACs | 208–211 | 800–933 | 440–493 | |

We therefore recommend prioritising B-based ACs as an economical breakthrough product, and an alternative to the high-cost commercial ACs for the purification of municipal or textile industrial wastewaters.

Data accessibility. Figures and tables files were downloaded with the main article.
Authors' contributions. The corresponding author and co-authors shared in the preparation of this research work.
Competing interests. We have no competing interests.
Funding. Funding is self-employed from all authors.

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
