## [Reviewer comments · Royal Society Open Science]

Review History

RSOS-190579.R0 (Original submission)

Review form: Reviewer 1 (Bogdan Tofanica)

Is the manuscript scientifically sound in its present form?

Yes

Are the interpretations and conclusions justified by the results?

No

Is the language acceptable?

Yes

Is it clear how to access all supporting data?

No

Do you have any ethical concerns with this paper?

No

Have you any concerns about statistical analyses in this paper?

No

Recommendation?

Reject

Comments to the Author(s)

Thank for the opportunity to review the article "Role of Pulping Process As Synergistic Treatment on Performance of Agro-based Activated Carbons" which addresses a fashionable research topic, relevant and important.

I appreciate the experimental work in the paper, but I do not feel that this research throw new light that deserves publication in a journal devoted to the development of scientific knowledge.

My major concerns regarding the content of the article are related to:

- the originality and novelty of the manuscript - the main objectives of the manuscript, producing AC from agricultural wastes has already been developed in many patents, research articles (DOI: 10.1016/j.rser.2015.02.051, 10.1016/j.jhazmat.2008.12.114, 10.1016/j.biortech.2007.07.058, 10.1016/j.biortech.2007.11.064, 10.1021/ie3012853, 10.1016/j.biortech.2009.02.028), and the energy balance can be a good method to check for the efficiency of the method;
- the scientific reliability of the work - the research results are predictable since there are a lot of available reports on the topic and the subject is well known;
- the methods used for the characterization of the materials reported and the validity of the work are not appropriate to the experiments, instead of references 33-35, I suggest TAPPI Methods, ISO standards, or established method from the literature; for the sorption properties, I recommend DOI: 10.4324/9780203762264, 10.1016/C2016-0-01974-5.

In conclusion, the novelty, the methodology and the results of the manuscript are quite common. The merits for publication are limited; therefore, the manuscript cannot meet the high quality of RSOS, a top-tier journal, dedicated to original research in science.

Review form: Reviewer 2 (Balal Yousaf)

Is the manuscript scientifically sound in its present form?

Yes

Are the interpretations and conclusions justified by the results?

No

Is the language acceptable?

Yes

Is it clear how to access all supporting data?

Not Applicable

Do you have any ethical concerns with this paper?

No

Have you any concerns about statistical analyses in this paper?

No

Recommendation?

Accept with minor revision (please list in comments)

Comments to the Author(s)

Basta et al., studied the role of pulping process as synergistic treatment on performance of agro-based activated carbons. The adsorption performance of the different prepared ACs was evaluated in terms of Iodine Numbers and their sorption properties for removing the methylene blue from aqueous solutions. Overall, this is a well-written manuscript with comprehensive and novel results. I suggest minor revision with following additions:

(1) Please note that Tables 2-7 contain measured data with significant digits which are not possible in field and even laboratory measurements. The data were apparently truncated with a spreadsheet. Please reduce the number of significant digits in these tables and possibly in the main text based on the accuracy and precision of the measurements. Please consult with an expert or search online (e.g., Google) on what significant digit means. Please note that significant digits are not equivalent to decimal places except that there is only zero before the decimal point.

(2) Please also take in account some important publications related to thermochemical conversion of agro-based feed-stocks to value-added products.

<https://doi.org/10.1016/j.jclepro.2018.05.246>

<https://doi.org/10.1016/j.jclepro.2018.05.034>

<https://doi.org/10.1016/j.jaap.2018.06.018>

Review form: Reviewer 3

Is the manuscript scientifically sound in its present form?

Yes

Are the interpretations and conclusions justified by the results?

Yes

Is the language acceptable?

Yes

Is it clear how to access all supporting data?

Not Applicable

Do you have any ethical concerns with this paper?

No

Have you any concerns about statistical analyses in this paper?

No

Recommendation?

Major revision is needed (please make suggestions in comments)

Comments to the Author(s)

This manuscript presented an experimental study to determine the effect of three pulping pretreatment processes on the property and performance of activated carbon produced from locally available rice straw (RS) and sugar cane bagasse (SCB). Preparation of agro-based activated carbon has been a hot research topic and has been extensively evaluated in previous studies. Although incremental, there were some novel aspects in the present work regarding the combination of raw materials and pretreatment processes. Thus, this study would add values to the research field. However, the authors should significantly revise and expand their discussion in the context of relevant previous studies, both in the Introduction and Results and discussion

sections. For instance, the Results and discussion section reads more like a simple description of the experimental results, and the authors did not sufficiently discuss their results with previous studies; the novelty of the present work needs to be highlighted more as well.

Decision letter (RSOS-190579.R0)

22-May-2019

Dear Professor Basta:

Title: Role of Pulping Process As Synergistic Treatment on Performance of Agro-based Activated Carbons

Manuscript ID: RSOS-190579

The editor assigned to your manuscript has now received comments from reviewers. We would like you to revise your paper in accordance with the referee and Subject Editor suggestions which can be found below (not including confidential reports to the Editor). Please note this decision does not guarantee eventual acceptance.

Please submit your revised paper before 14-Jun-2019. Please note that the revision deadline will expire at 00.00am on this date. If we do not hear from you within this time then it will be assumed that the paper has been withdrawn. In exceptional circumstances, extensions may be possible if agreed with the Editorial Office in advance. We do not allow multiple rounds of revision so we urge you to make every effort to fully address all of the comments at this stage. If deemed necessary by the Editors, your manuscript will be sent back to one or more of the original reviewers for assessment. If the original reviewers are not available we may invite new reviewers.

Please also include the following statements alongside the other end statements. As we cannot publish your manuscript without these end statements included, if you feel that a given heading is not relevant to your paper, please nevertheless include the heading and explicitly state that it is not relevant to your work.

- Acknowledgements

• Funding statement

Please include a funding section after your main text which lists the source of funding for each author.

RSC Associate Editor:
Comments to the Author:
(There are no comments.)

RSC Subject Editor:
Comments to the Author:
(There are no comments.)

Reviewers' Comments to Author:
Reviewer: 1

Comments to the Author(s)

Thank for the opportunity to review the article "Role of Pulping Process As Synergistic Treatment on Performance of Agro-based Activated Carbons" which addresses a fashionable research topic, relevant and important.

I appreciate the experimental work in the paper, but I do not feel that this research throw new light that deserves publication in a journal devoted to the development of scientific knowledge.

My major concerns regarding the content of the article are related to:

- the originality and novelty of the manuscript - the main objectives of the manuscript, producing AC from agricultural wastes has already been developed in many patents, research articles (DOI: 10.1016/j.rser.2015.02.051, 10.1016/j.jhazmat.2008.12.114, 10.1016/j.biortech.2007.07.058, 10.1016/j.biortech.2007.11.064, 10.1021/ie3012853, 10.1016/j.biortech.2009.02.028), and the energy balance can be a good method to check for the efficiency of the method;
- the scientific reliability of the work - the research results are predictable since there are a lot of available reports on the topic and the subject is well known;
- the methods used for the characterization of the materials reported and the validity of the work are not appropriate to the experiments, instead of references 33-35, I suggest TAPPI Methods, ISO

standards, or established method from the literature; for the sorption properties, I recommend DOI: 10.4324/9780203762264, 10.1016/C2016-0-01974-5.

In conclusion, the novelty, the methodology and the results of the manuscript are quite common. The merits for publication are limited; therefore, the manuscript cannot meet the high quality of RSOS, a top-tier journal, dedicated to original research in science.

Reviewer: 2

Comments to the Author(s)

Basta et al., studied the role of pulping process as synergistic treatment on performance of agro-based activated carbons. The adsorption performance of the different prepared ACs was evaluated in terms of Iodine Numbers and their sorption properties for removing the methylene blue from aqueous solutions. Overall, this is a well-written manuscript with comprehensive and novel results. I suggest minor revision with following additions:

(1) Please note that Tables 2-7 contain measured data with significant digits which are not possible in field and even laboratory measurements. The data were apparently truncated with a spreadsheet. Please reduce the number of significant digits in these tables and possibly in the main text based on the accuracy and precision of the measurements. Please consult with an expert or search online (e.g., Google) on what significant digit means. Please note that significant digits are not equivalent to decimal places except that there is only zero before the decimal point.

(2) Please also take in account some important publications related to thermochemical conversion of agro-based feed-stocks to value-added products.

<https://doi.org/10.1016/j.jclepro.2018.05.246>

<https://doi.org/10.1016/j.jclepro.2018.05.034>

<https://doi.org/10.1016/j.jaap.2018.06.018>

Reviewer: 3

Comments to the Author(s)

This manuscript presented an experimental study to determine the effect of three pulping pretreatment processes on the property and performance of activated carbon produced from locally available rice straw (RS) and sugar cane bagasse (SCB). Preparation of agro-based activated carbon has been a hot research topic and has been extensively evaluated in previous studies. Although incremental, there were some novel aspects in the present work regarding the combination of raw materials and pretreatment processes. Thus, this study would add values to the research field. However, the authors should significantly revise and expand their discussion in the context of relevant previous studies, both in the Introduction and Results and discussion sections. For instance, the Results and discussion section reads more like a simple description of the experimental results, and the authors did not sufficiently discuss their results with previous studies; the novelty of the present work needs to be highlighted more as well.

Author's Response to Decision Letter for (RSOS-190579.R0)

See Appendix A.

RSOS-190579.R1 (Revision)

Review form: Reviewer 2 (Balal Yousaf)

Is the manuscript scientifically sound in its present form?

Yes

Are the interpretations and conclusions justified by the results?

Yes

Is the language acceptable?

Yes

Do you have any ethical concerns with this paper?

No

Recommendation?

Accept as is

Comments to the Author(s)

Accept

Review form: Reviewer 3

Is the manuscript scientifically sound in its present form?

Yes

Are the interpretations and conclusions justified by the results?

Yes

Is the language acceptable?

Yes

Do you have any ethical concerns with this paper?

No

Recommendation?

Accept as is

Comments to the Author(s)

The authors have addressed the issues brought by the reviewers, and this reviewer does not have any additional comments.

Decision letter (RSOS-190579.R1)

25-Jun-2019

Dear Professor Basta:

Title: Role of Pulping Process As Synergistic Treatment on Performance of Agro-based Activated Carbons

Manuscript ID: RSOS-190579.R1

It is a pleasure to accept your manuscript in its current form for publication in Royal Society Open Science. The chemistry content of Royal Society Open Science is published in collaboration with the Royal Society of Chemistry.

RSC Associate Editor: 1
Comments to the Author:
(There are no comments.)

RSC Associate Editor: 2
Comments to the Author:
(There are no comments.)

Reviewer(s)' Comments to Author:

Reviewer: 2

Comments to the Author(s)

Accept

Reviewer: 3

Comments to the Author(s)

The authors have addressed the issues brought by the reviewers, and this reviewer does not have any additional comments.

Appendix A

Dear Editor Prof. Anthony Stace and Associate Editor Dr Ya-Wen Wang

We would like to express our gratitude for the Editor and Reviewers efforts to improve the quality of this manuscript. We have tried our best to address all issues indicated in the report. In the revised version, we have highlighted the changes to our manuscript using the grey color. Here, we would like to address the Reviewer's concerns as follows:

Reviewer 1:

Thank for the opportunity to review the article “Role of Pulping Process As Synergistic Treatment on Performance of Agro-based Activated Carbons” which addresses a fashionable research topic, relevant and important.

I appreciate the experimental work in the paper, but I do not feel that this research throw new light that deserves publication in a journal devoted to the development of scientific knowledge.

My major concerns regarding the content of the article are related to:

- the originality and novelty of the manuscript - the main objectives of the manuscript, producing AC from agricultural wastes has already been developed in many patents, research articles (DOI: 10.1016/j.rser.2015.02.051, 10.1016/j.jhazmat.2008.12.114, 10.1016/j.biortech.2007.07.058, 10.1016/j.biortech.2007.11.064, 10.1021/ie3012853, 10.1016/j.biortech.2009.02.028), and the energy balance can be a good method to check for the efficiency of the method;
- the scientific reliability of the work - the research results are predictable since there are a lot of available reports on the topic and the subject is well known;
- the methods used for the characterization of the materials reported and the validity of the work are not appropriate to the experiments, instead of references 33-35, I suggest TAPPI Methods, ISO standards, or established method from the literature; for the sorption properties, I recommend DOI: 10.4324/9780203762264, 10.1016/C2016-0-01974-5.

In conclusion, the novelty, the methodology and the results of the manuscript are quite common. The merits for publication are limited; therefore, the manuscript cannot meet the high quality of RSOS, a top-tier journal, dedicated to original research in science.

Answer

Thanks for the comments and your time

* With regards to Reviewer’s view about the novelty, we agree well with the viewpoint that many articles reported on the utilizing the agricultural wastes in production of activated carbons, but the work in the submitted article does not focus on using available agricultural wastes as precursors for ACs **but it deals with changing the constituents of RS & SCB via applying different pulping processes (alkali, sulphite and neutral sulphite), and evaluating their role on quality of the produced AC.** The changes in chemical constituents were estimated by many techniques available to us (chemical analyses, elemental analyses, thermal analysis, FT-IR spectra). In the manuscript, we illustrated that changes in the pulping process is effective on the particular sorption properties of RS and SCB based ACs. Interestingly, pulping process is a profound modification of the SCB based fibers, where it induced a clear increase of the specific surface areas of the corresponding ACs even through their impact on the sorption of methylene blue and iodine.

Moreover, the promising achievements are clear from the adsorption data we submitted in this paper, superior to the literature reported data dealing with agro-based ACs using H₃PO₄ as activator (we summarized it in Table 8).

* with regard to Reviewer comment "- the methods used for the characterization of the materials reported and the validity of the work are not appropriate to the experiments, instead of references 33-35, I suggest TAPPI Methods, ISO standards, or established method from the

literature; for the sorption properties, I recommend DOI: 10.4324/9780203762264, 10.1016/C2016-0-01974-5"

Answer

The Standard tests to evaluate the adsorption capacity of any carbon material are focused on studying its surface area, as well as its adsorption capacity to methylene blue and iodine value. These tests were conducted as an indicator to the efficiency of carbon materials to remove the contaminants from liquids, especially dyes. These evaluations are reported in many literatures. Moreover, these tests were performed in such a way that we can compare the obtained results with the previous performed ACs from RS and SCB pulps, which were evaluated by the very same tests (and available for us).

We have misunderstood the point raised by reviewer#1 who recommended articles such as :

DOI: 10.4324/9780203762264 & DOI: 10.1016/C2016-0-01974-5"

These chapters focus on "Potential of Biochar for Managing Metal Contaminated Areas, in Synergy With Phytomanagement or Other Management Options"

DOI: 10.1016/C2016-0-01974-5" is also a chapter dealing the potential for using biochar in phytomanagement and to control uptake of hazardous trace elements in agriculture. A final section focuses on the relatively limited field of experience that is available today.

We do not see the clear and direct relationship of these interesting publications with our experimental study. We decided not to include these references.

Reviewer: 2

Basta et al., studied the role of pulping process as synergistic treatment on performance of agro-based activated carbons. The adsorption performance of the different prepared ACs was evaluated in terms of Iodine Numbers and their sorption properties for removing the methylene blue from aqueous solutions. Overall, this is a well-written manuscript with comprehensive and novel results. I suggest minor revision with following additions:

Answer

Thank you about your recommendation which give us the opportunity to extend our publication in Royal Society Open Science.

(1) Please note that Tables 2-7 contain measured data with significant digits which are not possible in field and even laboratory measurements. The data were apparently truncated with a spreadsheet. Please reduce the number of significant digits in these tables and possibly in the main text based on the accuracy and precision of the measurements. Please consult with an expert or search online (e.g., Google) on what significant digit means. Please note that significant digits are not equivalent to decimal places except that there is only zero before the decimal point.

Answer : This comment is important as spreadsheet allows many digits induced by successive calculations, even though they do not have any physical meaning. In the Table caption, we have included that the accuracy of the measurements related in Table 2 is 0.01% for the elemental analyses and 0.1°C for TGA measurements. In Table 4, the IR characteristics MHBS and Cr.I. can be estimated with a precision of 0.01 whereas the textural analyses display the figures usually reported in many other publications. In Tables 6 and 7, the r^2 parameter is provided with the usual four figures, precise enough to discriminate between different models. We therefore modified the data in the manuscript accordingly and we apologize for having provided raw spreadsheet data.

(2) Please also take in account some important publications related to thermochemical conversion of agro-based feed-stocks to value-added products.

<https://doi.org/10.1016/j.jclepro.2018.05.246>

<https://doi.org/10.1016/j.jclepro.2018.05.034>
<https://doi.org/10.1016/j.jaap.2018.06.018>

Answer

Thank you for your valuable suggestion. We provided these References in the revised version of the manuscript [Refs. 15-17].

Reviewer: 3

This manuscript presented an experimental study to determine the effect of three pulping pretreatment processes on the property and performance of activated carbon produced from locally available rice straw (RS) and sugar cane bagasse (SCB). Preparation of agro-based activated carbon has been a hot research topic and has been extensively evaluated in previous studies. Although incremental, there were some novel aspects in the present work regarding the combination of raw materials and pretreatment processes. Thus, this study would add values to the research field. However, the authors should significantly revise and expand their discussion in the context of relevant previous studies, both in the Introduction and Results and discussion sections. For instance, the Results and discussion section reads more like a simple description of the experimental results, and the authors did not sufficiently discuss their results with previous studies; the novelty of the present work needs to be highlighted more as well.

Answer

Thank you so much for your suggestion and your efforts to improve our submitted article, which give us the opportunity to extend our publication in Royal Society Open Science.

With regard to your opinion "the authors should significantly revise and expand their discussion in the context of relevant previous studies, both in the Introduction and Results and discussion sections.

We would like to clarify that, we focused on evaluating the quality of produced activated carbons versus the changes in chemical analyses, which were performed via different pulping processes. This study was carried out to recommend the pulping process on agro-waste which will be promising in carbon production. Further evaluation was carried out via comparing our finding data with the literature reported data dealing with agro-based ACs. We gathered these comparisons in Table 8.

Finally, thanks again for all efforts by editorial staff and reviewers, and we hope that the revised version of our manuscript (LORD willing) meets the high standard of your Journal.

v. best, may our LORD bless you about your time
Altaf H. Basta (Res.Prof.)
NRC, Cairo, Egypt